# SemanticNVS: Improving Semantic Scene Understanding in Generative Novel View Synthesis

**Xinya Chen** [1]  **Christopher Wewer** [1]  **Jiahao Xie** [1]  **Xinting Hu** [1]  **Jan Eric Lenssen** [1]
**Project Page**: https://semanticnvs.github.io/

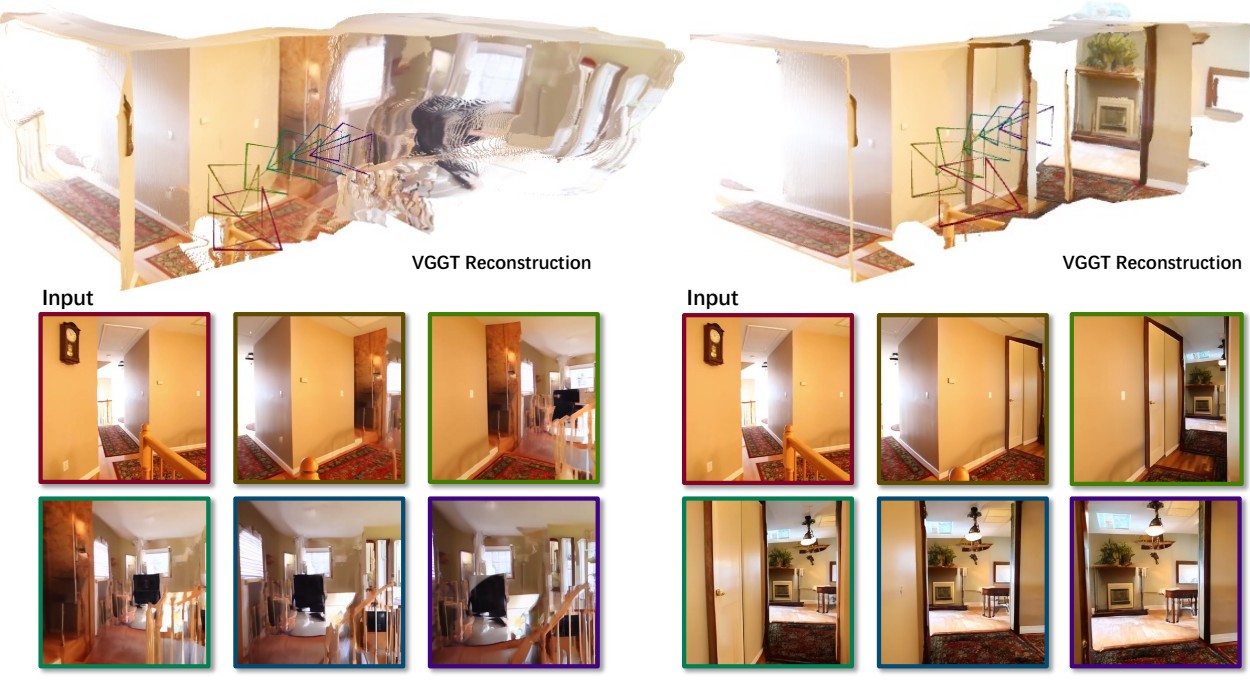

*Figure 1.* Integrating pre-trained semantic feature extractors clearly improves generative novel view synthesis quality in camera-conditioned diffusion models. Through better understanding of conditioning, our `SemanticNVS` is able to generate more consistent and high quality scenes (right), compared to a baseline without such mechanisms (left, SEVA (Zhou et al., 2025)), which shows unrealistic generation for content farer away from the input conditioning. The improvement is clearly visible in generated views (bottom) as well as subsequent VGGT (Wang et al., 2025) reconstructions (top). Border colors encode distance to the input view.

## Abstract

We present `SemanticNVS`, a camera-conditioned multi-view diffusion model for novel view synthesis (NVS), which improves generation quality and consistency by integrating pre-trained semantic feature extractors. Existing NVS methods perform well for views near the input view, however, they tend to generate semantically implausible and distorted images

under long-range camera motion, revealing severe degradation. We speculate that this degradation is due to current models failing to fully understand their conditioning or intermediate generated scene content. Thus, we propose to integrate pre-trained semantic feature extractors to incorporate stronger scene semantics as conditioning to achieve high-quality generation even at distant viewpoints. We investigate two different strategies, (1) warped semantic features and (2) an alternating scheme of *understanding* and *generation* at each denoising step. Experimental results on multiple datasets demonstrate the clear qualitative and quantitative (4.69%-15.26% in FID) improvement over state-of-the-art alternatives.

[1]Max Planck Institute for Informatics, Saarland Informatics Campus, Germany. Correspondence to: Jan Eric Lenssen <jlenssen@mpi-inf.mpg.de>.

# 1. Introduction

Generative novel view synthesis is emerging as a key technique for applications in entertainment, robotics, and 3D reconstruction. Given a single input view and a target camera trajectory, the goal is to synthesize realistic novel views that follow the given camera trajectory.

Recent methods typically adopt multi-view or video diffusion models (Zhou et al., 2025; Wan et al., 2025) and condition generation on camera poses via Plücker ray maps (Zhou et al., 2025), warped images (Yu et al., 2025b), or their combination (Cao et al., 2025). While effective in regions well observed by the input, these approaches often degrade in unobserved areas, producing semantically implausible and distorted content when the camera moves away from the input view. In this work, we improve the quality and semantic consistency of such generated videos by integrating pre-trained semantic features for stronger conditioning.

Finding the underlying reasons for semantically implausible generations and hallucinations in diffusion models is an important open research problem. However, recent research (Zhang et al., 2023; Wewer et al., 2025) suggests that stronger conditioning, which narrows down the space of possible generations and, therefore, reduces the distribution complexity, leads to samples with higher quality and consistency. Intuitively, consider the example of an image showing parts of a kitchen. If the model fully understands the high-level semantics contained in the image, for example, the existence of a stove, sink, and oven, it might deduce that the rest of the room should probably follow a kitchen-like layout, including appropriate furniture, such as a table or cupboards. The distribution to sample from becomes narrower and potentially simpler to model.

Our key hypothesis is that existing methods are not fully capable of capturing and leveraging such semantics in the given conditioning. Especially, conditioning signals such as warped inputs are incomplete due to limited overlap with the input (Fig. 2), making it challenging for the denoising network to understand object identity and semantics. Moreover, diffusion models denoise from noisy intermediate states where semantic cues are corrupted, increasing the difficulty of understanding the content that the model is generating.

Building on this hypothesis, we investigate how we can leverage pre-trained foundation models to better understand input and intermediate generated content, and, as a consequence, improve generative novel view synthesis in long-range camera motion. We present `SemanticNVS`, a method that integrates DINOv2 (Caron et al., 2021) into multi-view diffusion models to improve semantic understanding before generation. Concretely, we propose and investigate two different strategies: (1) warping semantic features from existing views into novel views to enable the

semantic understanding of input content, and (2) an alternating scheme of *semantic understanding* and *generation* that extracts semantic features of intermediate clean samples at each inverse diffusion step and uses them as conditioning for the next generation step, providing richer semantic cues than the noisy input alone.

In exhaustive experiments on multiple datasets, we demonstrate that both of the proposed techniques for intermediate scene and image understanding clearly improve the generation results qualitatively (as shown in Fig. 1, 4, 3), quantitatively by 4.69%-15.26% in FID, and 28.77%-30.00% in image-quality drift for long trajectory generation.

In summary, our contributions are:

- We find that current video generators do not leverage existing conditioning to the full amount and that further improvements in semantic scene and image understanding can also improve generative models for NVS.

- We introduce a mechanism that conditions NVS generation models on extracted and geometrically warped semantic features from existing conditioning views.

- We propose a novel alternating scheme of *understanding* and *generation* that leverages pre-trained feature extractors to improve conditioning in between individual diffusion steps.

# 2. Related Work

Generative novel view synthesis (NVS) aims to synthesize unseen views from as little as a single input view and a target camera trajectory, and has surged with large-scale latent diffusion models. Due to the scarcity of 3D data, recent methods typically adapt pretrained text-to-image/video diffusion models using iterative warp-and-inpaint (Seo et al., 2024), camera conditioning (He et al., 2025), or additional pretrained priors (Wu et al., 2026).

**Warp & Inpaint.** One line of work performs NVS by warping the input view to target views and inpainting missing regions using pre-trained diffusion models (Müller et al., 2024; Tang et al., 2023; Ren & Wang, 2022). SceneScape (Fridman et al., 2023) and LucidDreamer (Chung et al., 2025) further couple this idea with a growing 3D representation, alternating between novel view rendering, diffusion-based inpainting, and backprojection into 3D. ViewCrafter (Yu et al., 2025b), See3D (Ma et al., 2025), and FlexWorld (Chen et al., 2025) extend this paradigm to video diffusion models and recent 3D reconstruction models. Several works strengthen geometric control by warping coordinate maps or diffusion noise (Seo et al., 2024; Gu et al., 2025; Burgert et al., 2025). While effective for incremental scene expansion, these pipelines typically rely on partial and fragmented warped observations as the primary conditioning signal, making it challenging to maintain high-level semantic co-

herence under large viewpoint changes or long trajectories.

**Camera Conditioning.** Another line of research generates target views by conditioning diffusion models directly on camera information. MotionCtrl (Wang et al., 2024), CameraCtrl (He et al., 2025), Uni3C (Cao et al., 2025), and CamCtrl3D (Popov et al., 2025) explore encodings of extrinsic and intrinsic camera parameters, (Plücker) ray maps, and combinations with warped RGB images to control the camera trajectory of video diffusion models. In parallel, CAT3D (Gao et al., 2024) and SEVA (Zhou et al., 2025) turn image generators into multi-view models conditioned on ray maps of desired target cameras. By training on multiple real-world scene datasets and applying a two-stage inference of widespread anchor and in-between novel views, they enable NVS under longer trajectories and large viewpoint changes. Despite these advances, achieving both high visual quality and stable semantics remains difficult when the camera moves far from the observed view: geometric conditions alone may not sufficiently constrain the space of plausible completions in unobserved regions.

**Leveraging Pre-trained Priors for NVS.** To improve 3D consistency in generative NVS, concurrent works (Wu et al., 2026) propose to leverage features or network components of pre-trained geometric priors like VGGT (Wang et al., 2025). Related to that, another line of research (Szymanowicz et al., 2025; Huang et al., 2025; Zhang et al., 2025b; Zhu et al., 2025; Liu et al., 2025) leverages depth as an auxiliary supervision signal to improve the consistency of novel view synthesis. However, while semantic priors like DINO (Oquab et al., 2023) have been successfully leveraged in image generation, e.g., as supervision for intermediate features of diffusion models as proposed by REPA (Yu et al., 2025a), their use in the context of NVS remains unexplored. This is where SemanticNVS comes into play. By leveraging pre-trained semantic priors, we improve scene understanding and, as a result, enable higher-quality and semantically consistent novel views synthesis.

# 3. SemanticNVS

In this section, we present SemanticNVS. We first review diffusion preliminaries in Sec. 3.1, then give an architecture overview in Sec. 3.2. Finally, we describe our two semantic-conditioning components in Sec. 3.3 and Sec. 3.4.

## 3.1. Preliminaries

We study generative novel view synthesis (NVS) in a conditional diffusion framework. Given one or a few source views with known camera parameters and a target camera trajectory as the condition, the goal is to synthesize a sequence of novel views that follows the trajectory.

The conditional diffusion model comprises a forward nois-

ing process and a learned reverse-time denoising process. We follow the EDM (Karras et al., 2022) formulation with a continuous noise level $\sigma(t)$, $t \in [0, 1]$. The forward process perturbs clean data by additive Gaussian noise:

$$x_t = x_0 + \sigma(t)\,\epsilon, \qquad \epsilon \sim \mathcal{N}(0, I), \tag{1}$$

where $\sigma(t) \in [\sigma_{\min}, \sigma_{\max}]$.

The reverse process is parameterized by a network that predicts the noise $\epsilon_\theta(x_t, t, c)$ conditioned on the noisy input, its noise level, and conditioning signals $c$. From the predicted noise, we form a one-step denoised estimate:

$$\hat{x}_0^t = x_t - \sigma(t)\,\epsilon_\theta(x_t, t, c). \tag{2}$$

At inference time, we use a discretized noise schedule $\{\sigma_t\}_{t=0}^T$ with $\sigma_0 = \sigma_{\max}$ and $\sigma_T = \sigma_{\min}$, and denoise from larger to smaller noise levels. Concretely, we replace $\sigma(t)$ in Eq. 2 by $\sigma_t$ at each discretized step. An Euler step from $x_t$ to $x_{t-1}$ is:

$$\begin{aligned} x_{t-1} &= \hat{x}_0^t + \sigma_{t-1}\,\epsilon_\theta(x_t, t, c) \\ &= x_t + (\sigma_{t-1} - \sigma_t)\,\epsilon_\theta(x_t, t, c). \end{aligned} \tag{3}$$

In practice, diffusion generation is typically performed in the latent space of a pretrained autoencoder. For brevity, we use $x$ to denote diffusion states throughout the paper.

## 3.2. Overall Architecture

Fig. 2 provides an overview of SemanticNVS. We build our framework on top of SEVA (Zhou et al., 2025), a camera-conditioned diffusion backbone for novel view synthesis.

SEVA performs conditional generation by providing the denoising network with camera conditioning signals $c$ Plücker ray maps. We further incorporate warped RGB for better performance, following Cao et al. (2025); Popov et al. (2025). However, these conditions cannot provide enough high-level semantics for visually realistic and semantically plausible novel-view synthesis (especially under long-range camera motion). We therefore augment the conditioning with pretrained semantic features (i.e., DINO features), to provide richer semantic evidence while keeping the same diffusion backbone.

Specifically, we introduce two complementary semantic augmentations on top of the camera-conditioned diffusion framework: (i) we geometrically warp pretrained semantic features extracted from the source views into the target view, yielding warped semantic features ($\mathbf{F}_w$ in Fig. 2) that anchor semantics in visible regions (Sec. 3.3); and (ii) we alternate semantic understanding and generation during denoising, i.e., extracting semantic features from the current one-step denoised estimate $\hat{x}_0^t$ and feeding them back as step-wise conditioning for the next update $t \to t-1$, which improves

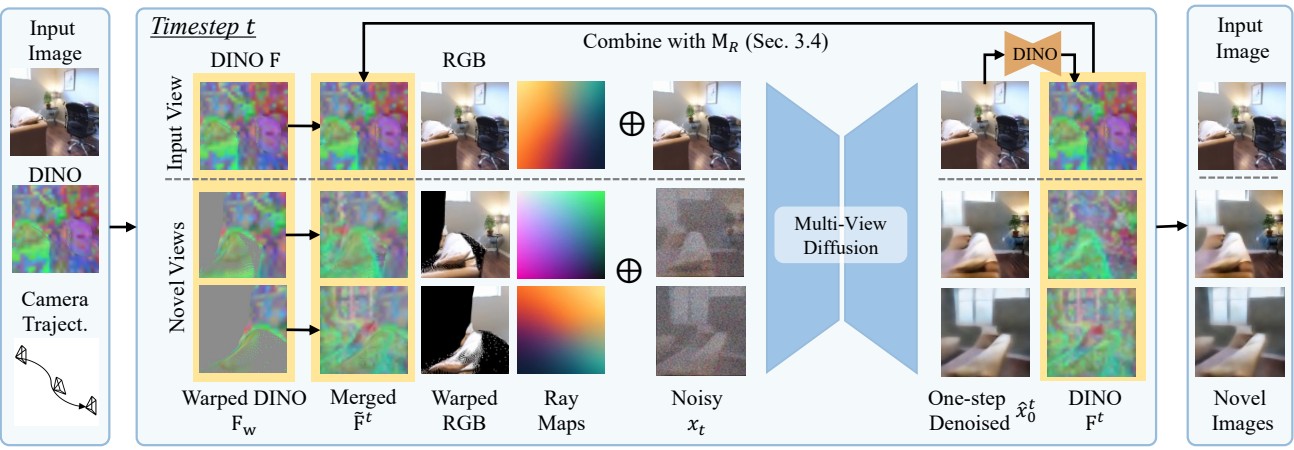

*Figure 2.* **Method Overview.** `SemanticNVS` integrates semantic DINO features into the multi-view diffusion setup in two different ways. First, it provides warped features from the given input view and uses them as additional conditioning. Second, it extracts DINO features from intermediate generations $\hat{x}_0^t$ of the previous iteration and uses them to complete the warped DINO features.

semantic consistency in unobserved regions and along long trajectories (Sec. 3.4). For clarity, we describe the method for a single target frame $x$ in the main text. Full-trajectory generation follows by applying the same procedure to all target frames in parallel with frame-dependent step indices.

### 3.3. Warped Semantic Features

Warped RGB observations are often incomplete due to occlusions and limited source coverage, resulting in fragmented appearance evidence in target views. These partial observations make it difficult for the denoising network to infer object identity and semantics, especially in unobserved regions of novel views. To address this issue, we introduce semantic conditioning that provides high-level cues to help the model recognize objects under severe missing regions.

Specifically, given an input image $\mathbf{I} \in \mathbb{R}^{H \times W \times 3}$, we extract semantic features $\mathbf{F} \in \mathbb{R}^{H \times W \times C}$ using a DINO encoder

$$\mathbf{F} = f_{\text{DINO}}(\mathbf{I}). \tag{4}$$

Following the warping strategy in Yu et al. (2025b), we employ a dense stereo model (e.g., VGGT (Wang et al., 2025)) to reconstruct a point cloud from the input view. We then project the per-point DINO features into target cameras along a given camera trajectory, rendering warped semantic feature images $\mathbf{F}_w \in \mathbb{R}^{H \times W \times C}$. These warped semantic features complement warped RGB by providing robust object-level context even when appearance is incomplete.

Since the semantic features are high-dimensional, we employ a lightweight linear projection to reduce their channel dimensionality before using them as conditioning. To improve training stability and performance, we first $\ell_2$-normalize the semantic features along the channel dimension before projecting them to a compact representation

with a $1 \times 1$ convolution $\phi(\cdot)$

$$\tilde{\mathbf{F}}_w = \phi\left(\frac{\mathbf{F}_w}{\|\mathbf{F}_w\|_2}\right), \quad \tilde{\mathbf{F}}_w \in \mathbb{R}^{H \times W \times C'}, \quad (C' \ll C). \tag{5}$$

Finally, we feed $\tilde{\mathbf{F}}_w$ as an additional conditioning signal to the denoising U-Net, alongside the other camera-related conditions. In contrast to the warped RGB, we do not additionally encode the DINO features into a spatially smaller latent space using the VAE encoder.

### 3.4. Semantic Features from Intermediate Samples

During sampling, the network predicts an estimate of the clean sample $\hat{x}_0^t$ at each denoising step $t$, and the sampler then re-injects noise to obtain $x_{t-1}$, which serves as the input to the next step (Eq. 3). Since $x_{t-1}$ remains a noisy input, it contains limited semantic cues and is therefore difficult for the network to interpret.

To improve semantic awareness during generation, we introduce an explicit understanding signal at every denoising step. While $x_{t-1}$ is corrupted, the intermediate prediction $\hat{x}_0^t$ is noise-free and thus more amenable to semantic feature extraction. We therefore leverage a foundational vision model to derive dense semantics from $\hat{x}_0^t$. Specifically, we extract DINO features

$$\mathbf{F}^t = f_{\text{DINO}}(\hat{x}_0^t). \tag{6}$$

Moreover, the warped DINO features $\mathbf{F}_w$ are rendered from the observed input views and are thus more reliable in regions supported by input evidence. Accordingly, we use $\mathbf{F}_w$ in rendered regions and fall back to $\mathbf{F}^t$ elsewhere. Concretely, we fuse them using the rendering mask $\mathbf{M}_R$:

$$\tilde{\mathbf{F}}^t = \mathbf{M}_R \odot \mathbf{F}_w + (1 - \mathbf{M}_R) \odot \mathbf{F}^t, \tag{7}$$

where $\odot$ denotes element-wise multiplication and $\mathbf{M}_R$ indicates pixels covered by rendering from the input views. This

alternating scheme of understanding and denoising provides semantic guidance for the sampling process at every step.

**Blur for Approximating $\hat{x}_0^t$ in Training.** The conditioning on semantic features in every denoising step leverages the intermediate samples $\hat{x}_0^t$ at inference time. However, during training we do not have access to paired data $(\hat{x}_0^t, x_0)$. Empirically, we observe that $\hat{x}_0^t$ is typically blurred. Therefore, we apply a blur operator, i.e., a Gaussian filter to $x_0$ to approximate $\hat{x}_0^t$ and use the resulting image as a surrogate. With increasing noise level $t$, we observe that the predicted clean sample $\hat{x}_0^t$ during sampling becomes increasingly blurred. Hence, we mimic this effect by increasing the blur strength of the Gaussian filter with the timestep $t$. We provide further details in the appendix.

## 4. Experiments

In this section, we describe our experimental evaluation of `SemanticNVS`. After introducing the experimental setup in Sec. 4.1 and implementation details in Sec. 4.2, we provide qualitative and quantitative comparisons with baselines in Sec. 4.3, showing the advantage of our method by incorporating scene understanding in generative NVS. Sec. 4.4 ablates on the introduced semantic conditionings, the choice of the pre-trained feature extractor, and further provides a comparison with REPA. Please refer to additional details and qualitative results including videos in the **appendix** and the **project page**.

### 4.1. Experimental Setup

**Datasets.** We evaluate NVS on two real-world datasets, RealEstate10K (Zhou et al., 2018) and Tanks-and-Temples (Knapitsch et al., 2017), which cover indoor and outdoor scenes. Tanks-and-Temples is out of the training distribution and used to evaluate generalization. We consider two trajectory regimes: *short trajectory* contains 80-100 frames and *long trajectory* contains more than 250 frames. For each, we randomly sample 100 videos for evaluation.

**Baselines.** We consider three baselines: ViewCrafter (Yu et al., 2025b), Uni3C (Cao et al., 2025), and SEVA (Zhou et al., 2025). ViewCrafter and Uni3C are limited to single-window generation with a maximum number of frames and do not provide a mechanism to roll the window forward (i.e., sliding-window/iterative generation with context propagation). As a result, they cannot synthesize long trajectories beyond the supported window length. Therefore, we uniformly subsample the test trajectories with a fixed stride to 20 frames for a fair comparison. In contrast, SEVA supports full-trajectory generation in a two-stage procedure of anchor and interpolated view generation. Thus, we provide an additional comparison with SEVA on the full trajectories.

**Metrics.** We evaluate novel view synthesis quality based on six factors: **(1) Distribution fidelity.** We measure distribution-level fidelity by computing Fréchet Inception Distance (FID) (Heusel et al., 2017) between generated frames and the ground truth test set. **(2) Image quality.** We assess per-frame perceptual quality using the *Imaging Quality* metric from VBench (Huang et al., 2024). We report the dataset-level score by averaging over all test videos. **(3) Image-quality drift along the trajectory.** To quantify degradation along the camera trajectory, we adopt the drifting measurement from Zhang et al. (2025a) and compute the start–end contrast as an image-quality metric. **(4) Camera control accuracy**. To evaluate the accuracy of camera control, we assess the alignment between ground truth camera poses and poses estimated from the generated images, following Yu et al. (2025b). **(5) Cross-view detail preservation**. Following previous works (Zhou et al., 2025; Yu et al., 2025b), we measure how faithfully the generated views match the ground-truth target views, using PSNR, SSIM (Wang et al., 2004), and LPIPS (Zhang et al., 2018). **(6) 3D consistency.** We evaluate geometric consistency across generated views using MEt3R (Asim et al., 2025). Since MEt3R is defined for a pair of images, we compute it on temporally adjacent views along the generated trajectory.

### 4.2. Implementation Details

**Training Details.** We build our method on top of SEVA (Zhou et al., 2025) and initialize from its released pretrained checkpoint. Our models are trained on a mixed real-world corpus composed of RealEstate10K (Zhou et al., 2018) and DL3DV (Ling et al., 2024), which jointly cover both indoor and outdoor scenes and comprise approximately 75k videos. Training is conducted on 4 GPUs with a batch size of 1 per GPU using AdamW, with a learning rate of $1.25 \times 10^{-5}$ and weight decay of $1 \times 10^{-2}$. The final model is trained for 60,000 iterations. We enable the proposed *Semantic Features from Intermediate Samples* training paradigm after 37,000 iterations; empirically, this staged schedule yields better performance than applying the paradigm from the beginning of training. Unless otherwise specified, we use image resolution $H = W = 576$, latent resolution $h = w = 72$, projected semantic feature dimension $C' = 32$, and window size $N = 21$.

**Inference Details.** Following SEVA (Zhou et al., 2025), we adopt a two-stage inference procedure. In the first stage, we synthesize a set of *anchor* views uniformly sampled along the target camera trajectory. In the second stage, we generate the intermediate views by interpolating between adjacent anchors. `SemanticNVS` significantly improves the results of the first stage. Since the second stage mostly performs interpolation of previously generated anchor views, it does not profit as much from additional semantic conditioning.

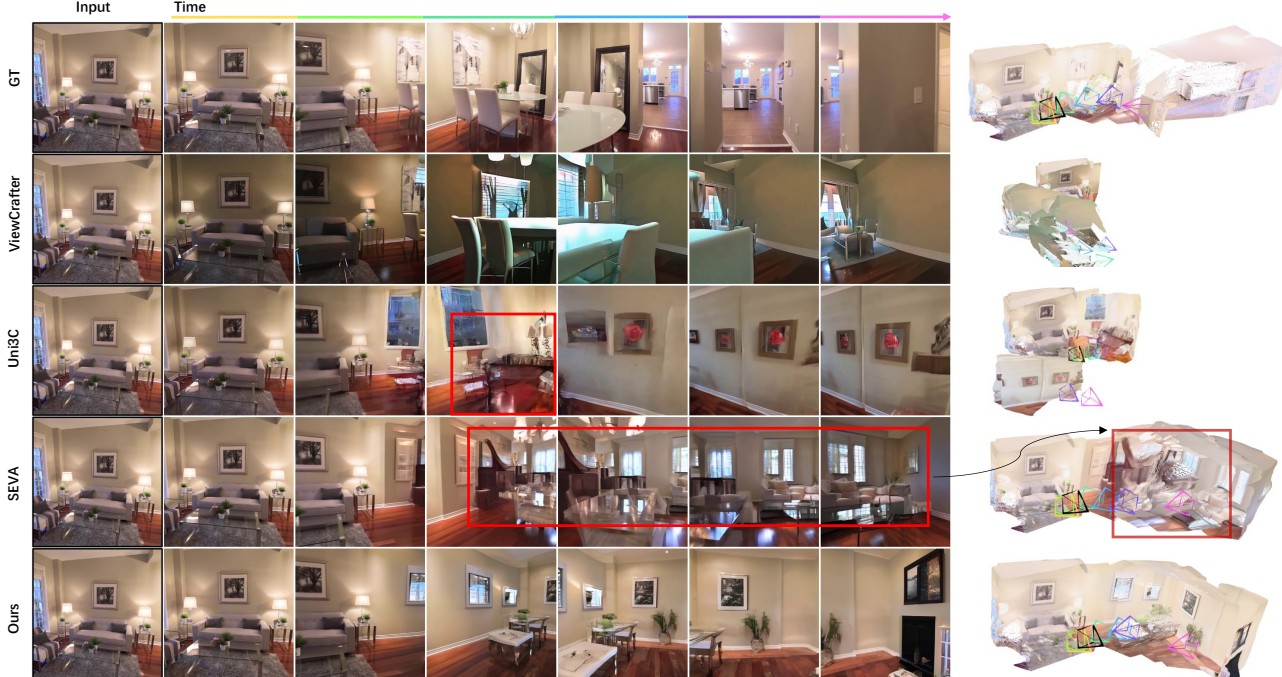

*Figure 3.* **Qualitative Comparison on RealEstate10K.** ViewCrafter and Uni3C struggle to follow the long camera trajectory accurately. SEVA produces degraded views when moving far from the input. In contrast, our method better adheres to the target trajectory, generates more realistic novel views, and yields a more coherent underlying geometry when reconstructing the 3D scene from the generated frames. More results can be found in the **appendix** and the **project page**.

| Method | Short Trajectory (80∼100) | | | | | | | | | Long Trajectory (≥250) | | | | | | | | |
|---|---|---|---|---|---|---|---|---|---|---|---|---|---|---|---|---|---|---|
| | FID ↓ | ImQ ↑ | Drift ↓ | RE ↓ | TE ↓ | MEt3R ↓ | PSNR ↑ | SSIM ↑ | LPIPS ↓ | FID ↓ | ImQ ↑ | Drift ↓ | RE ↓ | TE ↓ | MEt3R ↓ | PSNR ↑ | SSIM ↑ | LPIPS ↓ |
| ViewCrafter | 32.929 | 58.499 | 0.387 | 0.329 | 0.030 | 0.075 | 15.000 | 0.559 | 0.356 | 37.703 | 58.599 | 0.597 | 4.612 | 0.339 | 0.162 | 11.021 | 0.458 | 0.595 |
| Uni3C | 25.308 | 55.246 | 0.335 | 0.358 | 0.028 | 0.069 | 16.208 | 0.599 | 0.323 | 39.572 | 49.984 | 0.533 | 3.434 | 0.197 | 0.151 | 12.652 | 0.521 | 0.564 |
| SEVA | 26.376 | 59.145 | 0.357 | 0.202 | **0.014** | 0.067 | 15.517 | 0.579 | 0.357 | 31.344 | 57.971 | 0.511 | 0.535 | 0.055 | 0.110 | 12.871 | 0.517 | 0.519 |
| Ours | **22.726** | **62.060** | **0.251** | **0.189** | 0.019 | **0.067** | **18.024** | **0.646** | **0.251** | **26.561** | **66.458** | **0.364** | **0.468** | **0.054** | **0.109** | **13.817** | **0.542** | **0.450** |

*Table 1.* **Quantitative Comparison on RealEstate10K.** SemanticNVS achieves the best overall performance across a variety of metrics, showing significant improvements in generation quality, geometric consistency, and reconstruction quality.

### 4.3. Comparison to Baselines

**Qualitative Comparison.** Fig. 3 and Fig. 4 present qualitative comparisons on *long* camera trajectories for RealEstate10K and Tanks-and-Temples. For each sequence, we uniformly sample six generated frames along the trajectory ordered left-to-right by time. To further assess the plausibility of the generated geometry and camera control accuracy, we feed the generated frames into VGGT to obtain a reconstructed scene together with the estimated camera poses. The top color bar and the camera frustums share the same time-based color coding, with the input highlighted in black.

When the camera moves far away from the input view, ViewCrafter and Uni3C fail to follow the target camera trajectory, leading to severe viewpoint drift and content collapse. As a consequence, the reconstructions exhibit broken geometry and misaligned estimated poses. SEVA follows the camera trajectory more accurately, but produces unrealistic and blurry content for extrapolated views far from the input, as shown in the highlighted regions. As a result,

the reconstructed 3D scene suffers from noisy geometry. In contrast, our method maintains both trajectory adherence and visual realism across large viewpoint changes.

**Quantitative Comparison.** We report the quantitative evaluation of baselines and our method on RealEstate10K in Table 1, on Tanks-and-Temples in Table 2. On RealEstate10K, we improve the generation results quantitatively by 10.20%-15.26% in FID, 4.93%-13.41% in image quality, and 25.07%-28.77% in image-quality drift. The pronounced reduction in drift indicates that our generations remain stable even when the camera moves far away from the input view, where the target views have little overlap with the input. On Tanks-and-Temples, which is out of the training distribution and thus serves as a generalization benchmark, we improve FID by 4.69%–14.98% and image-quality drift by 1.83%–30.00%. While ViewCrafter attains competitive ImQ and Drift, it exhibits worse camera control and 3D consistency, especially on long trajectories. SEVA shows a clear degradation even on short trajectories on this dataset. Overall, our method achieves the best overall performance, demonstrating good generalization.

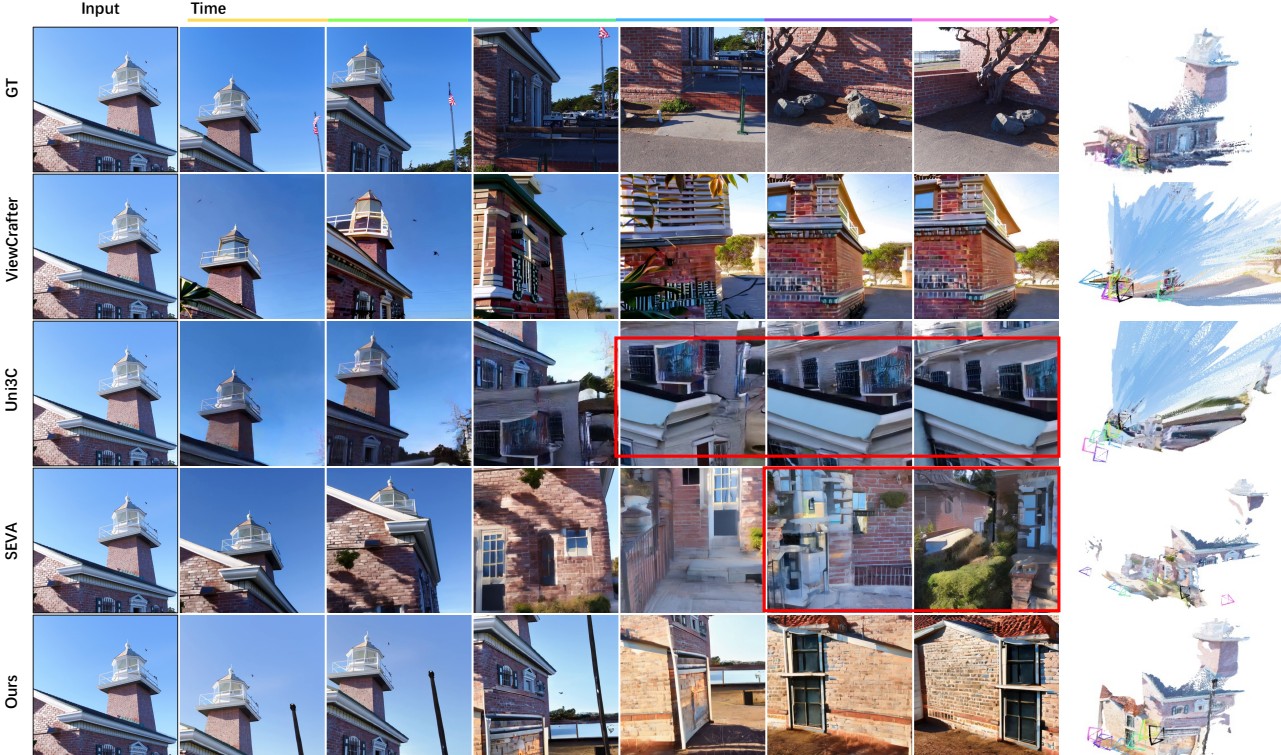

*Figure 4.* **Qualitative Comparison on Tanks-and-Temples.** ViewCrafter, Uni3C, and SEVA fail to follow the long camera trajectory and produce unrealistic or degraded views. In contrast, our method better adheres to the target trajectory, generates more realistic novel views, and yields a more coherent underlying geometry when reconstructing from the generated frames.

| Method | Short Trajectory (80∼100) | | | | | | | | | Long Trajectory (≥250) | | | | | | | | |
|---|---|---|---|---|---|---|---|---|---|---|---|---|---|---|---|---|---|---|
| | FID↓ | ImQ↑ | Drift↓ | RE↓ | TE↓ | MEt3R↓ | PSNR↑ | SSIM↑ | LPIPS↓ | FID↓ | ImQ↑ | Drift↓ | RE↓ | TE↓ | MEt3R↓ | PSNR↑ | SSIM↑ | LPIPS↓ |
| ViewCrafter | 38.587 | **71.029** | **0.215** | 0.901 | 0.008 | 0.104 | 14.557 | 0.427 | 0.364 | 57.998 | 68.867 | 0.380 | 5.001 | 0.027 | 0.187 | 12.137 | 0.352 | 0.510 |
| Uni3C | 36.166 | 61.491 | 0.366 | 0.401 | 0.006 | 0.088 | 15.560 | 0.464 | 0.356 | 57.873 | 55.189 | 0.473 | 2.564 | 0.020 | 0.152 | 13.402 | 0.409 | 0.505 |
| SEVA | 44.749 | 68.344 | 0.340 | 0.381 | **0.002** | 0.098 | 12.978 | 0.400 | 0.474 | 46.564 | 66.258 | 0.436 | 0.640 | **0.005** | 0.127 | 12.196 | 0.373 | 0.531 |
| Ours | **30.749** | 70.985 | 0.219 | **0.197** | 0.003 | **0.085** | **16.240** | **0.495** | **0.275** | **44.381** | **69.829** | **0.266** | **0.577** | 0.008 | **0.119** | **13.825** | **0.426** | **0.405** |

*Table 2.* **Quantitative Comparison on Tanks-and-Temples.** While SEVA performs significantly worse on scenes out of training distribution, `SemanticNVS` is more robust and achieves the best overall performance on Tanks-and-Temples.

| Method | Full Long Trajectory (≥250) | | | | | | | | |
|---|---|---|---|---|---|---|---|---|---|
| | FID↓ | ImQ↑ | Drift↓ | RE(°)↓ | TE (m)↓ | MEt3R↓ | PSNR↓ | SSIM↓ | LPIPS↓ |
| SEVA | 25.347 | 57.816 | 0.554 | **0.297** | 0.035 | 0.070 | 13.310 | 0.529 | 0.502 |
| Ours | **21.842** | **64.463** | **0.359** | 0.310 | **0.032** | 0.070 | **14.155** | **0.553** | **0.443** |

*Table 3.* **Full Trajectory Comparison on RealEstate10K.** Applying the two-stage anchor generation and view interpolation for both methods, we beat SEVA in almost all metrics.

We further compare against SEVA on full-trajectory generation in Table 3, where our method achieves superior performance, improving FID, image quality, and image-quality drift by 13.83%, 11.50%, and 35.20%, respectively, while maintaining comparable camera control and 3D consistency.

**4.4. Ablation Studies**

**Ablation on Core Components.** Table 4 ablates the *Warped Semantic Features* and *Semantic Features from Intermediate Samples*, which we denote as *Warped DINO* and *Iterative DINO*. Since SEVA does not release its training code and

| Method | Long Trajectory (≥250) | | | | | | | | |
|---|---|---|---|---|---|---|---|---|---|
| | FID↓ | ImQ↑ | Drift↓ | RE↓ | TE↓ | MEt3R↓ | PSNR↑ | SSIM↑ | LPIPS↓ |
| SEVA | 31.344 | 57.971 | 0.511 | 0.535 | 0.055 | 0.110 | 12.871 | 0.517 | 0.519 |
| SEVA finetuned | 31.896 | 56.794 | 0.603 | 0.539 | 0.076 | 0.114 | 12.609 | 0.515 | 0.556 |
| + Warped RGB | 29.942 | 59.201 | 0.499 | 0.518 | 0.063 | 0.124 | 13.790 | 0.529 | 0.465 |
| + Warped DINO | 28.908 | 60.452 | 0.424 | 0.506 | **0.052** | 0.111 | 13.809 | 0.538 | 0.452 |
| + Iterative RGB | 28.260 | 62.820 | 0.399 | 0.472 | 0.056 | 0.110 | 13.815 | 0.539 | 0.451 |
| + Iterative DINO | **26.561** | **66.458** | **0.364** | **0.468** | 0.054 | **0.109** | **13.817** | **0.542** | **0.450** |

*Table 4.* **Ablation on Core Components on RealEstate10K.** Warped and Iterative DINO both individually improve the overall performance. Using DINO features of intermediate samples (Iterative DINO) outperforms the equivalent Iterative RGB variant.

training data, we additionally report results of SEVA finetuned with our training pipeline and data for a fair comparison. Following Uni3C, we augment SEVA with *Warped RGB*, which serves as our baseline.

Adding *Warped DINO* consistently improves all metrics. Further introducing *Iterative DINO* yields additional gains, with particularly substantial improvements in FID, ImQ, and Drift, while achieving competitive translation error (TE). Qualitative results show the same trend in Fig. 5.

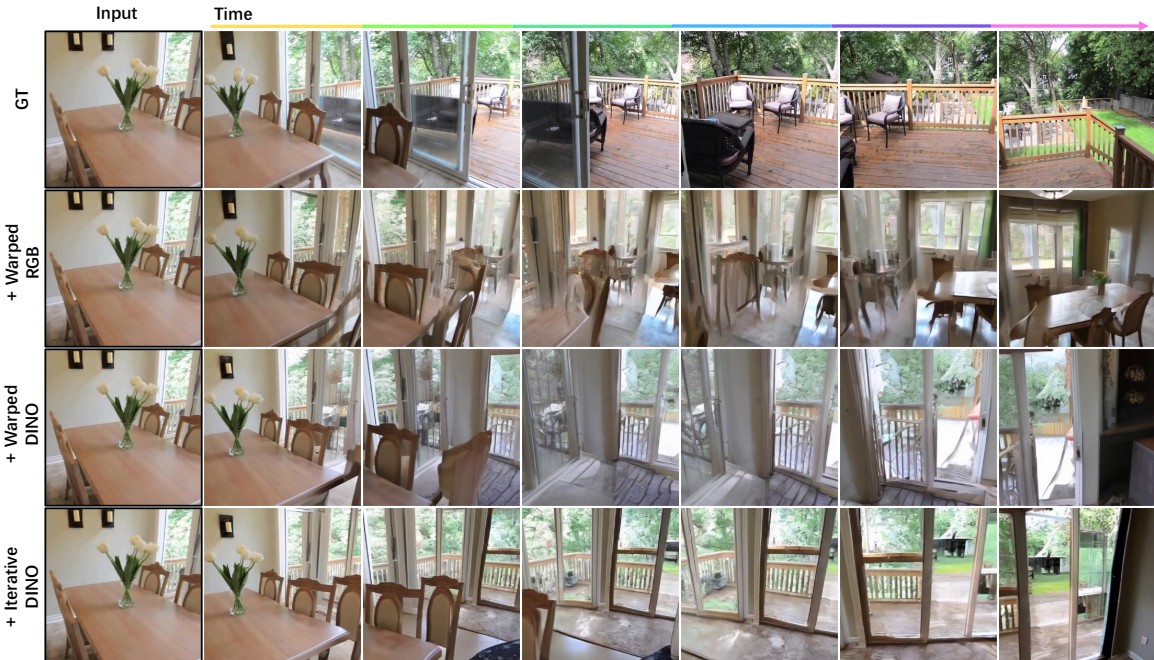

*Figure 5.* **Qualitative Ablation on RealEstate10K.** Warped RGB fails to generate the floor-to-ceiling glass window on the right side of the input view. Adding Warped DINO produces a clear window, but the chair remains incomplete. Further guiding sampling with intermediate-sample features (Iterative DINO) enables view-consistent synthesis of the entire scene.

| Method | Long Trajectory ($\geq$250) | | | | | | | | |
|---|---|---|---|---|---|---|---|---|---|
| | FID↓ | ImQ↑ | Drift↓ | RE↓ | TE↓ | MEt3R↓ | PSNR↑ | SSIM↑ | LPIPS↓ |
| DINOv3 | 29.058 | 64.590 | 0.416 | **0.458** | 0.055 | 0.112 | 13.544 | **0.549** | 0.456 |
| VGGT | 28.704 | 61.781 | 0.407 | 0.538 | 0.059 | 0.109 | 13.608 | 0.540 | 0.452 |
| DINOv2 | **26.561** | **66.458** | **0.364** | 0.468 | **0.054** | **0.109** | **13.817** | 0.542 | **0.450** |

*Table 5.* **Feature Ablation on RealEstate10K.** DINOv2 achieves the best overall performance among different foundation models.

| Method | Long Trajectory ($\geq$250) | | | | | | | | |
|---|---|---|---|---|---|---|---|---|---|
| | FID↓ | ImQ↑ | Drift↓ | RE↓ | TE↓ | MEt3R↓ | PSNR↑ | SSIM↑ | LPIPS↓ |
| SEVA finetuned | 31.896 | 56.794 | 0.603 | 0.539 | 0.076 | 0.114 | 12.609 | 0.515 | 0.556 |
| + REPA | 30.579 | 63.231 | 0.384 | 0.721 | 0.067 | 0.116 | 12.787 | 0.503 | 0.538 |
| + Ours | **26.561** | **66.458** | **0.364** | **0.468** | **0.054** | **0.109** | **13.817** | **0.542** | **0.450** |

*Table 6.* **Comparison with REPA on RealEstate10K.** While both making use of DINO as semantic prior, SemanticNVS outperforms REPA for generative NVS.

We also ablate a variant that, at each denoising step, conditions only on $\hat{x}_0^t$ rather than its DINO feature. We denote this variant as *Iterative RGB*. This variant remains beneficial, but the improvement is smaller than using DINO features, suggesting that conditioning on the noise-free estimate $\hat{x}_0^t$ provides more informative guidance than using $x_t$ alone. Moreover, conditioning on the DINO feature of $\hat{x}_0^t$ injects stronger semantic understanding, further strengthening the conditioning and leading to the best performance.

**Comparison with REPA.** We compare with REPA (Yu et al., 2025a), which distills DINO features into the intermediate representations of the diffusion model. While usually applied for single-image generation, we adapt it to our setting of generative NVS with a multi-view diffusion model by supervising the intermediate features of all jointly generated views during training with the DINO alignment loss.

As shown in Table 6, REPA improves performance over the baseline, but remains inferior to our approach. We attribute this gap to the way semantic understanding is incorporated: our method provides *explicit* understanding by decoupling semantic interpretation from the generative model, whereas REPA injects semantics *implicitly* into the diffusion backbone, which can consume generation capacity and weaken the model's ability to focus on synthesis.

**Comparison with Different Features.** We further compare different semantic features, including DINOv2, DINOv3, and VGGT features in Table 5. DINOv3 yields clear improvements in ImQ and Drift, but does not improve FID. Using VGGT features improves all metrics and achieves competitive camera control. Among the three, DINOv2 provides the most consistent gains across all metrics.

## 5. Conclusion

We presented SemanticNVS, a method that integrates pretrained semantic feature extractors into multi-view diffusion to strengthen the semantic conditioning of the model. Our experiments demonstrate significant qualitative and quantitative improvements in semantic consistency and generation quality, especially in weak conditioning domains, e.g., long trajectory generation. The finding suggests that information extraction from conditioning signals in current multi-view diffusion models still has potential for improvements and that further advances in self-supervised pre-training can benefit generative novel view synthesis as well.

# Acknowledgements

This project was partially funded by the Saarland/Intel Joint Program on the Future of Graphics and Media. Jan Eric Lenssen is supported by the German Research Foundation (DFG) - 556415750 (Emmy Noether Programme, project: Spatial Modeling and Reasoning).

# Impact Statement

This paper presents work whose goal is to advance the field of Machine Learning. There are many potential societal consequences of our work, none of which we feel must be specifically highlighted here.

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

# Appendix

In this supplementary material, we provide additional implementation details (Section A), evaluation protocols (Section B), experimental analyses (Section C, Section D, Section E, Section F, Section G, Section H), and limitations (Section I) to complement the main paper.

## A. Implementation Details

This section describes how we approximate the predicted clean sample $\hat{x}_0^t$ during training, and how we construct a timestep-dependent surrogate for extracting DINO features.

We apply a blur operator (i.e., Gaussian blur) to $x_0$ to approximate $\hat{x}_0^t$ and use the resulting features as a surrogate. We observe that the predicted clean sample $\hat{x}_0^t$ becomes increasingly blurred as the diffusion timestep $t$ increases. To approximate this effect during training, we set the blur strength of the Gaussian blur operator grows with the timestep $t$.

Concretely, we define the blur standard deviation as:

$$\tau_t \;=\; \tau_{\min} + \frac{t}{T}\,(\tau_{\max} - \tau_{\min}). \qquad \text{(A1)}$$

Following standard practice, we choose an odd kernel size proportional to $\tau_t$:

$$k_t \;=\; 2\,\text{round}(3\tau_t) + 1. \qquad \text{(A2)}$$

We then obtain the surrogate of $x_0$ via:

$$\tilde{x}_0^{(t)} \;=\; \mathcal{G}(x_0; \tau_t, k_t), \qquad \text{(A3)}$$

where $\mathcal{G}(\cdot; \tau, k)$ denotes Gaussian blur with standard deviation $\tau$ and kernel size $k$. Finally, we extract DINO features from $\tilde{x}_0^{(t)}$ and use them as a surrogate for the DINO features of $\hat{x}_0^t$. We set the minimum blur standard deviation $\tau_{\min} = 0.1$, and the maximum blur standard deviation $\tau_{\max} = 30$.

## B. Metrics

This section introduces the detailed evaluation metrics used in our experiments, including metrics to evaluate image quality, image-quality drift along the trajectory, and the camera control accuracy.

**Image quality.** We assess per-frame perceptual quality using the *Imaging Quality* metric from VBench (Huang et al., 2024). Given a generated video $V = \{I_t\}_{t=1}^{T}$, we follow the official VBench evaluation pipeline to compute a frame-wise quality score $q(I_t)$ and aggregate it over time.

$$M(V) = \frac{1}{T}\sum_{t=1}^{T} q(I_t), \qquad \text{(A4)}$$

where $q(\cdot)$ denotes the VBench *Imaging Quality* evaluator applied to each frame. We report the dataset-level score by averaging $M(V)$ over all test videos.

**Image-quality drift along the trajectory.** To quantify degradation along the camera trajectory, we adopt the drifting measurement in Zhang et al. (2025a) and compute the start–end contrast for an image-quality metric $M$ (i.e.,*Imaging Quality*):

$$\Delta_{\text{drift}}^{M}(V) = |M(V_{\text{start}}) - M(V_{\text{end}})|, \qquad \text{(A5)}$$

where $V_{\text{start}} = \{I_t\}_{t=1}^{\lfloor 0.15T \rfloor}$ and $V_{\text{end}} = \{I_t\}_{t=\lceil 0.85T \rceil}^{T}$ denote the first and last $15\%$ frames of $V$, respectively.

**Camera control accuracy**. To evaluate the accuracy of camera control, we assess the alignment between the camera poses of generated images and the ground truth camera poses, following Yu et al. (2025b). Specifically, we use VGGT (Wang et al., 2025) to extract poses from generated views. Then we transfer camera poses relative to the first frame and normalize translation by the furthest frame. We calculate the rotation error RE by comparing ground truth and extracted rotation matrices of each generated sequence in degrees:

$$\text{RE} = \arccos\left(\frac{1}{2}\left(\text{tr}\big(\mathbf{R}_{\text{gen}}\mathbf{R}_{\text{gt}}^{\top}\big) - 1\right)\right). \qquad \text{(A6)}$$

where R denotes a rotation matrix. We compute translation error TE in meters as:

$$\text{TE} \;=\; \|\mathbf{t}_{\text{gt}} - \mathbf{t}_{\text{gen}}\|_2. \qquad \text{(A7)}$$

## C. Comparison with Concurrent Work

In this section, we report comparisons with a concurrent camera-controlled generative baseline Gen3R (Huang et al., 2026) in Table A1 and Table A2. Our method achieves the best performance across both datasets and trajectory regimes, with particularly clear advantages on long trajectories and on the out-of-distribution Tanks-and-Temples dataset. These results provide additional evidence that our semantic conditioning design is effective.

## D. Source of End-to-End Gain

The end-to-end gain comes from improving the anchor views. We only apply the proposed semantic conditioning to anchor generation, and the improved anchor views subsequently lead to better interpolated views. To verify the source of improvement, we use our method only for anchor-view generation and replace the interpolation stage with SEVA. This yields similar performance, with FID of 21.90 versus 21.84 for the full model. This confirms that the observed end-to-end gain is primarily attributable to the improved anchor views.

| Method | Short Trajectory (80~100) | | | | | | | | | Long Trajectory (≥250) | | | | | | | | |
|---|---|---|---|---|---|---|---|---|---|---|---|---|---|---|---|---|---|---|
| | FID↓ | ImQ↑ | Drift↓ | RE↓ | TE↓ | MEt3R↓ | PSNR↑ | SSIM↑ | LPIPS↓ | FID↓ | ImQ↑ | Drift↓ | RE↓ | TE↓ | MEt3R↓ | PSNR↑ | SSIM↑ | LPIPS↓ |
| Gen3R | 27.121 | 49.850 | 0.368 | 0.315 | 0.035 | **0.065** | 16.125 | 0.583 | 0.368 | 52.283 | 43.773 | 0.472 | 1.708 | 0.212 | 0.128 | 12.004 | 0.497 | 0.628 |
| Ours | **22.726** | **62.060** | **0.251** | **0.189** | **0.019** | 0.067 | **18.024** | **0.646** | **0.251** | **26.561** | **66.458** | **0.364** | **0.468** | **0.054** | **0.109** | **13.817** | **0.542** | **0.450** |

*Table A1.* Quantitative Comparison on RealEstate10K.

| Method | Short Trajectory (80~100) | | | | | | | | | Long Trajectory (≥250) | | | | | | | | |
|---|---|---|---|---|---|---|---|---|---|---|---|---|---|---|---|---|---|---|
| | FID↓ | ImQ↑ | Drift↓ | RE↓ | TE↓ | MEt3R↓ | PSNR↑ | SSIM↑ | LPIPS↓ | FID↓ | ImQ↑ | Drift↓ | RE↓ | TE↓ | MEt3R↓ | PSNR↑ | SSIM↑ | LPIPS↓ |
| Gen3R | 39.379 | 58.612 | 0.299 | 0.405 | 0.013 | 0.085 | 14.030 | 0.427 | 0.462 | 68.809 | 52.247 | 0.405 | 1.784 | 0.033 | 0.147 | 12.131 | 0.381 | 0.588 |
| Ours | **30.749** | **70.985** | **0.219** | **0.197** | **0.003** | 0.085 | **16.240** | **0.495** | **0.275** | **44.381** | **69.829** | **0.266** | **0.577** | **0.008** | **0.119** | **13.825** | **0.426** | **0.405** |

*Table A2.* Quantitative Comparison on Tanks-and-Temples.

| Model | SEVA finetuned | Ours w/ DINO ViT-S | Ours w/ DINO ViT-B | Ours w/ DINO ViT-L |
|---|---|---|---|---|
| FID↓ | 31.896 | 28.483 | 27.516 | 26.561 |
| Infer Time (s/img) ↓ | 1.301 | 1.396 | 1.399 | 1.404 |

*Table A3.* Ablation on DINOv2 Encoder.

| Method | SEVA finetuned | + Warped RGB | + Warped DINO | + Iterative DINO |
|---|---|---|---|---|
| FID↓ | 31.896 | 29.942 | 28.908 | 26.561 |
| Infer Time (s/img) ↓ | 1.301 | 1.303 | 1.303 | 1.404 |

*Table A4.* Computation cost comparison.

## E. Ablation on DINOv2 Encoder

In this section, we ablate different DINOv2 encoders. We use the ViT-L in our method. Table A3 reports FID, average inference time for different DINOv2 encoders. Smaller encoders provide weaker gains, while inference time stay comparable. This is because DINO feature extraction is only used in the first-stage generation, so its reduced cost has limited impact on total inference time. Irrespective of the used DINOv2 ViT variant, our method outperforms the SEVA baseline significantly.

## F. Computation Cost

We report the average inference time in Table A4, measured on an H100 over 10 videos with more than 250 frames each. Compared with the baseline (+Warped RGB), warped DINO and iterative DINO deliver significant performance gains with similar runtime. The extra computation from DINO extraction and warping is limited because these operations are performed only during the first-stage generation.

## G. Qualitative Comparison

Fig. A1 present qualitative comparisons on *long* camera trajectories for RealEstate10K. For each sequence, we uniformly sample six generated frames along the trajectory ordered left-to-right by time. To further assess the plausibility of the generated geometry and camera control accuracy, we feed the generated frames into VGGT to obtain a reconstructed scene together with the estimated camera poses. The top color bar and the camera frustums share the same time-based color coding, with the input highlighted in black.

## H. Qualitative Video Comparison

In the **project page**, we show more qualitative video comparisons against baselines.

## I. Limitations

Our method may be limited by its reliance on a pre-trained feature extractor. For example, if DINOv2 fails to capture certain semantic objects, the generation quality may degrade. Besides, the model may fail on scenes involving humans, animals, or dynamic textures, which may stem from limitations in the scope of its training data.

Moreover, our current work focuses on static scenes with short ( < 100 frames) to medium-length ( 200+ frames) trajectories. Scalability to very large scenes, such as KITTI-style sequences with thousands of frames, as well as dynamic scenes, is an interesting and important future direction.

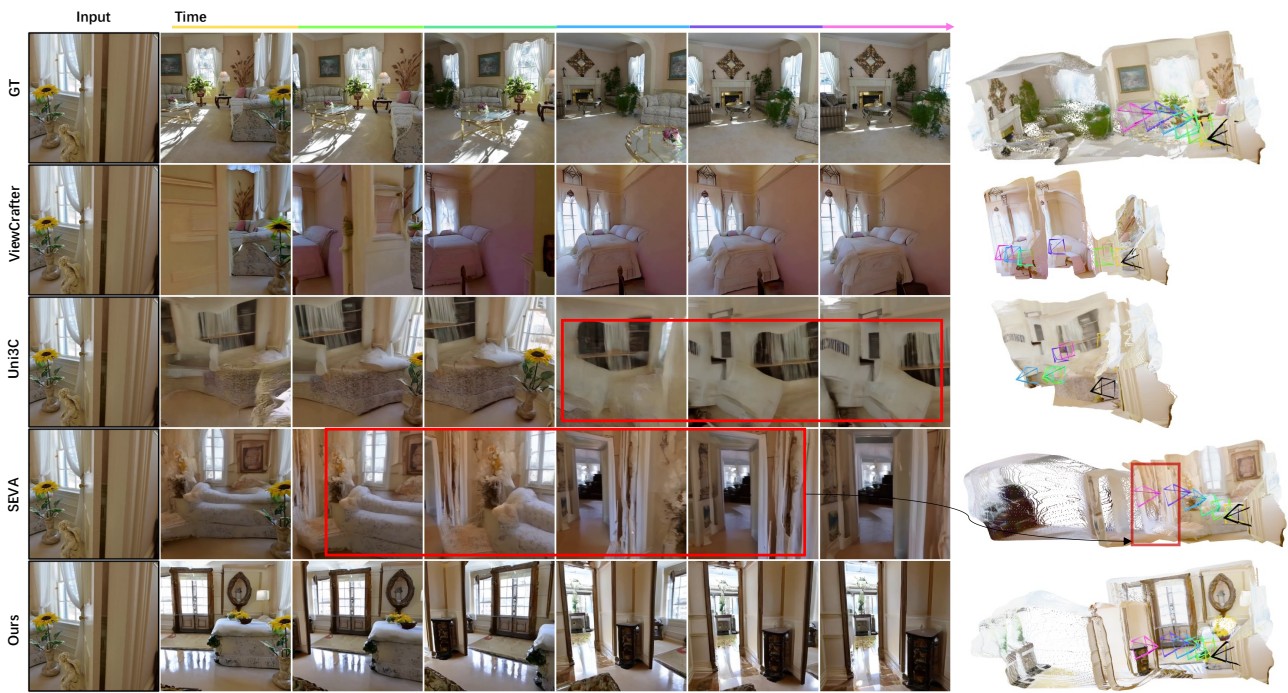

*Figure A1.* **Qualitative Comparison on RealEstate10K.**

