# OpenReview forum: "SemanticNVS: Improving Semantic Scene Understanding in Generative Novel View Synthesis"
_ICML.cc/2026/Conference — ICML 2026 regular_

### Official Review · Reviewer_P6Pt · 2026-03-12

**Soundness:** 3
**Presentation:** 3
**Significance:** 3
**Originality:** 3
**Overall Recommendation:** 4
**Confidence:** 4

**Summary:**

SemanticNVS addresses the degradation of image quality and consistency in novel view synthesis (NVS) when subjected to long-range camera motion. The authors argue that current models fail to leverage semantic understanding of the scene, relying instead on raw RGB inputs which are prone to occlusion and distortion. To bridge this, the authors integrate pre-trained semantic feature extractors (e.g., DINOv2) to provide stronger semantic conditioning. The framework explores two specific strategies: (1) warped semantic features to handle occlusion-induced data loss, and (2) an iterative alternating scheme that merges semantic features from intermediate denoising steps into the generative process. Experimental results show consistent improvements across multiple metrics (e.g., FID, LPIPS) compared to state-of-the-art baselines.

**Compliance With Llm Reviewing Policy:**

Affirmed.

**Final Justification:**

Thank you to the authors for the rebuttal, which has addressed all of my concerns. I will maintain my initial rating.

**Key Questions For Authors:**

see weakness

**Limitations:**

The authors should explicitly discuss the computational cost of the proposed pipeline, particularly the overhead of feature extraction and iterative merging. Additionally, they should acknowledge the potential limitation regarding the dependence on the pre-trained feature extractor (e.g., if DINOv2 fails to capture specific semantic objects, the generation will likely fail).

**Strengths And Weaknesses:**

### Strengths:
1. The motivation is highly compelling. Using semantic features (DINOv2) instead of RGB for warping is technically sound because semantic features are generally invariant to lighting and viewpoint changes, making them robust to occlusions and distortions during long-range motion.
2. The Iterative DINO strategy that merging semantic features from intermediate samples into the generative loop is a novel way to establish a feedback loop between the model's current "understanding" of the scene and the ongoing generation.
3. The empirical gains (as seen in Table 4) are significant and consistent across multiple metrics, indicating that the incorporation of semantic priors is a viable path for improving long-range temporal consistency in NVS.
### Weakness:
1. The paper lacks a discussion on the computational overhead. Integrating DINOv2 extraction, warping, and iterative merging adds significant inference-time complexity. The authors need to provide a latency/memory analysis to demonstrate the trade-off between quality gains and efficiency.
2. Table 5 shows DINOv3 underperforming compared to DINOv2. This is counter-intuitive as DINOv3 (or more recent variations) is generally expected to have richer representations. Without a discussion on why this occurred (e.g., resolution mismatch, overfitting, or domain shift), the contribution remains slightly opaque.
3. The inference workflow (whether it is sequential/autoregressive or fixed) is not described clearly. If the model is autoregressive, the accumulation of error is a major concern that needs to be addressed in the limitations.

---

> ### Author Rebuttal · Authors · 2026-03-30
>
> We thank all reviewers for their constructive comments.
> The reviewers remark that our method is novel (P6Pt), the motivation is highly compelling (P6Pt, KhQb, TY1i), the evaluation is broad (TY1i), the improvement is effective (P6Pt, KhQb, TY1i, DKvy), the gains are consistent across multiple settings and metrics (TY1i, P6Pt), the ablation is relevant (TY1i), the work addresses a meaningful limitation (KhQb, TY1i), and the paper is clearly written and easy to navigate (TY1i, DKvy).
> We now address the comments.
>
> **W1: Computational Overhead**
>
> We report the average inference time and peak memory in Table 1, measured on an H100 over 10 videos with more than 250 frames each. Compared with the baseline (+Warped RGB), warped DINO and iterative DINO deliver significant performance gains with similar runtime and only a minor increase from 11.466 GB to 12.290 GB in peak memory. The extra computation from DINO extraction and warping is limited because these operations are performed only during the first-stage generation. Their memory usage is temporary and does not accumulate across the pipeline. The small memory increase mainly comes from the DINO encoder and the higher diffusion-forward cost introduced by semantic conditioning.
>
> **Table 1.** Computation Cost Comparison.
> | Method | FID ↓  | Infer Time (s/img) ↓ | Mem (GB) ↓ |
> |---|:---:|:---:|:---:|
> | SEVA finetuned | 31.896 | 1.301 | 11.196 |
> | + Warped RGB | 29.942 | 1.303 | 11.466 |
> | + Warped DINO | 28.908 | 1.303 | 12.133 |
> | + Iterative DINO | 26.561 | 1.404 | 12.290 |
>
> ---
> **W2: Why DINOv2 is better?**
>
> While DINOv3 improves over DINOv2 on many dense semantic benchmarks, recent work suggests that DINOv2 can still be stronger for geometry- and correspondence-critical tasks. For example, Geo6DPose [1] reports that DINOv2 consistently outperforms DINOv3 across model scales for geometric matching, indicating stronger correspondence-preserving representations. This is also broadly consistent with PatchAlign3D [2], which uses DINOv2 as a strong geometric initialization and highlights the importance of preserving geometric coherence in dense features. We believe this is particularly relevant to generative NVS, where reliable cross-view correspondence is critical for maintaining multi-view consistency.
>
> [1] Geo6DPose: Fast Zero-Shot 6D Object Pose Estimation via Geometry-Filtered Feature Matching.
>
> [2] PatchAlign3D: Local Feature Alignment for Dense 3D Shape understanding.
>
> ---
> **W3: Inference Workflow**
>
> The input camera trajectory is the full sequence and fixed throughout inference. Our pipeline first generates anchor views and then interpolates the remaining views (L264–268). This procedure is not autoregressive.
>
> ---
> **Limitations**
>
> We thank the reviewer for the helpful suggestion. We will show the computational cost in the paper and discuss the dependence on the pre-trained feature extractor in the limitations.

---

> > ### Author Rebuttal · Reviewer_P6Pt · 2026-04-05
> >
> > The authors did a good job on addressing my concerns

---

> > > ### Author Response · Authors · 2026-04-07
> > >
> > > We thank the reviewer for the positive feedback and will incorporate the above discussions into the revised version. Since the concerns are resolved, we would also sincerely appreciate it if you would consider raising the score. Thank you again for the constructive feedback.

---

### Official Review · Reviewer_KhQb · 2026-03-12

**Soundness:** 3
**Presentation:** 3
**Significance:** 3
**Originality:** 3
**Overall Recommendation:** 4
**Confidence:** 4

**Summary:**

This paper presents **SemanticNVS**, a method designed to improve semantic consistency in novel view synthesis. The key idea is to introduce semantic-aware guidance during the reconstruction or rendering process so that the synthesized views better preserve high-level scene semantics across viewpoints. The method integrates semantic features into the scene representation and rendering pipeline, aiming to address the common issue that purely photometric supervision often fails to preserve semantic structures in novel views.

The authors evaluate their approach on multiple datasets and show that incorporating semantic information improves both perceptual quality and semantic consistency in rendered views. The results suggest that semantic guidance can help stabilize reconstruction and improve the interpretability of the generated views.

**Compliance With Llm Reviewing Policy:**

Affirmed.

**Key Questions For Authors:**

1. How sensitive is the method to the quality of semantic labels or segmentation predictions?
2. Does the approach generalize well to scenes with complex object interactions or heavy occlusions?
3. What is the computational overhead introduced by the semantic components compared to standard NVS pipelines?
4. Can the method handle open-vocabulary semantics or only predefined semantic classes?

**Limitations:**

- The approach may depend on reliable semantic annotations or segmentation models, which may not always be available.
- The scalability of the method to very large scenes or long video sequences is not fully explored.
- Performance in highly dynamic scenes or scenes with strong appearance changes remains unclear.

**Strengths And Weaknesses:**

## Strengths
- **Motivation is well justified.** The paper addresses an important limitation of many current novel view synthesis approaches: the lack of semantic consistency across viewpoints.
- **Semantic integration is meaningful.** Incorporating semantic information into the reconstruction pipeline is a promising direction that could improve robustness and scene understanding.
- **Improved semantic consistency.** The qualitative results suggest that the method produces views with better structural coherence compared with baseline methods.
- **Potential for broader applications.** The approach could be useful for tasks such as 3D scene understanding, robotics, or AR/VR applications where semantic awareness is important.

## Weaknesses
- **Limited novelty relative to existing semantic NVS approaches.** Several prior works have explored semantic guidance in neural rendering or NVS pipelines, and the paper could more clearly position its contribution relative to these methods.
- **Method description could be clearer.** Some parts of the pipeline, particularly how semantic features are integrated into the rendering process, would benefit from more detailed explanations.
- **Experimental comparisons are somewhat limited.** Including more recent baselines or stronger semantic-aware NVS methods would help better demonstrate the advantages of the approach.
- **Ablation studies could be expanded.** It would be useful to analyze how different components (e.g., semantic supervision strength or feature integration strategies) affect performance.

---

> ### Author Rebuttal · Authors · 2026-03-30
>
> We thank all reviewers for their constructive comments. The reviewers remark that our method is novel (P6Pt), the motivation is highly compelling (P6Pt, KhQb, TY1i), the evaluation is broad (TY1i), the improvement is effective (P6Pt, KhQb, TY1i, DKvy), the gains are consistent across multiple settings and metrics (TY1i, P6Pt), the ablation is relevant (TY1i), the work addresses a meaningful limitation (KhQb, TY1i), and the paper is clearly written and easy to navigate (TY1i, DKvy). We now address the comments.
>
> **W1: Novelty over Prior Semantic NVS**
>
> Our work focuses on generative NVS instead of neural rendering or reconstruction-based NVS pipelines. We are not aware of directly comparable prior work that explores the semantic conditioning in generative NVS. If the reviewer has specific prior works in mind, we would greatly appreciate the pointers and would be happy to discuss the distinctions in the second-round discussion.
>
> ---
> **W2: Method Description**
>
> We integrate semantic features by (1) incorporating warped semantic features as conditions to enable the semantic understanding of input content, and (2) proposing an alternating scheme of semantic understanding and generation that extracts the semantic features of intermediate clean samples at each inverse diffusion step and uses them as conditioning
> for the next generation step, providing richer semantic cues than the noisy input alone.
>
> ---
> **W3: Experimental Comparisons**
>
> We include a more recent baseline Gen3R (CVPR26) [1] in Table 1-4 of the reply of reviewer DKvy. We are not aware of directly comparable semantic-aware generative NVS methods. We would appreciate any specific pointers the reviewer may have in mind.
>
> [1] 3D Scene Generation Meets Feed-Forward Reconstruction
>
> ---
> **W4: Ablation Studies**
>
> We do not use semantic supervision in our method, so there is no semantic supervision strength to ablate.
>
> Regarding feature integration strategies, we provide ablations in Table 4 of the paper, including Warped DINO and Iterative DINO.
>
> ---
> **Q1: Sensitive to Semantic labels**
>
> Please note that our method does not leverage any ground truth semantic labels but only features extracted from pre-trained vision foundation models. Table 2 in the reply of reviewer TY1i shows that our approach outperforms baselines even with smaller DINOv2 variants and is therefore robust to the feature quality of weaker semantic priors.
>
> ---
> **Q2: Scenes with Occlusion**
>
> In the supplementary local website, we have provided examples for scenes with occlusion (e.g., the first case, which generates room regions behind windows or walls), where our method performs clearly better than the baselines.
>
> ---
> **Q3: Computational Overhead**
>
> We report the average inference time and peak memory in Table 1, measured on an H100 over 10 videos with more than 250 frames each. Compared with the baseline (+Warped RGB), warped DINO and iterative DINO deliver significant performance gains with similar runtime and only a minor increase from 11.466 GB to 12.290 GB in peak memory. The extra computation from DINO extraction and warping is limited because these operations are performed only during the first-stage generation. Their memory usage is temporary and does not accumulate across the pipeline. The small memory increase mainly comes from the DINO encoder and the higher diffusion-forward cost introduced by semantic conditioning.
>
> **Table 1.** Computation Cost Comparison.
> | Method | FID ↓ | Infer Time (s/img) ↓ | Mem (GB) ↓ |
> |---|:---:|:---:|:---:|
> | SEVA finetuned | 31.896 | 1.301 | 11.196 |
> | + Warped RGB | 29.942 | 1.303 | 11.466 |
> | + Warped DINO | 28.908 | 1.303 | 12.133 |
> | + Iterative DINO | 26.561 | 1.404 | 12.290 |
>
> ---
> **Q4: Open-vocabulary Semantics**
>
> Our semantic features are extracted using DINOv2, a self-supervised visual encoder trained on large-scale and diverse data. This allows our method to capture open-vocabulary semantic information rather than being restricted to a predefined set of semantic classes.
>
> ---
> **Very Large Scenes and Dynamic Scenes**
>
> Our current work focuses on static scenes with short ( <100 frames) to medium-length ( 200+ frames) trajectories, where we achieve state-of-the-art performance. Scalability to very large scenes, such as KITTI-style sequences with thousands of frames, as well as dynamic scenes, is an important but orthogonal direction that we leave for future work. We will clarify this scope in the paper.

---

> > ### Author Rebuttal · Reviewer_KhQb · 2026-04-07
> >
> > Thanks for the authors' response. I will keep my rating.

---

> > > ### Author Response · Authors · 2026-04-07
> > >
> > > We thank the reviewer for the positive feedback and will incorporate the above discussions into the revised version.

---

### Official Review · Reviewer_DKvy · 2026-03-13

**Soundness:** 3
**Presentation:** 3
**Significance:** 3
**Originality:** 3
**Overall Recommendation:** 4
**Confidence:** 4

**Summary:**

This paper focuses on a common problem for current NVS methods, typically based on camera-conditioned diffusion models, often struggle with spatial and semantic consistency when generating long camera trajectories. Specifically, common failures include distorted geometry or drifting content because the model lacks a deep understanding of the 3D scene's semantic structure from very limited input views. To mitigate the issue, the paper proposes integrating a pre-trained semantic feature extractor into the diffusion model to enhance the spatial understanding ability. In general, the core contribution of this work is demonstrating that high-level semantic understanding is a crucial ingredient for robust 3D scene generation. By explicitly conditioning on semantic features, SemanticNVS bridges the gap between 2D image synthesis and 3D-consistent world modeling.

**Compliance With Llm Reviewing Policy:**

Affirmed.

**Final Justification:**

Thanks for the response. I will keep my original score.

**Key Questions For Authors:**

I will adjust my score based on the response.

**Limitations:**

Please see the weakness part.

**Strengths And Weaknesses:**

Strength:
1. The paper provides a thorough discussion of current methodologies in NVS and the common challenges. By demonstrating the performance gains achieved through semantic integration, the authors effectively illustrate that explicitly conditioning on semantic features is crucial for high-quality NVS.
2. The paper is well-written and easy to follow. Both the core arguments and the experimental settings are presented with great clarity, making the technical content highly accessible.

Weakness:
1. Novelty of Methodology: Injecting semantic features to enhance the understanding ability of generative models is not a fundamentally new concept; similar approaches have been utilized in prior works (e.g., [1]). While I believe leveraging this characteristic specifically for NVS is a good practice, the technical novelty feels somewhat limited.
2. Comparison with Prompt-Based Conditioning: Modern diffusion models (e.g., Wan, HunyuanVideo) already exhibit strong prompt-following capabilities. If descriptive scene prompts are incorporated during the NVS training and inference process, it is likely that they already provide a high-level scene understanding. In such a context, how much additional gain does the explicit injection of semantic features provide? A comparison or discussion on this would clarify the necessity of the proposed module in the era of large-scale video diffusion models.
3. More baseline: For the baselines, I noticed that a substantial portion relies on warping-based pattern, which are inherently prone to geometric distortion and content drift. To make the results more solid, I suggest the authors include comparisons with more recent camera-controlled generative baselines that do not rely on warping. This would further validate the superiority of the proposed semantic conditioning mechanism.

[1] Yu S, Kwak S, Jang H, et al. Representation alignment for generation: Training diffusion transformers is easier than you think[J]. arXiv preprint arXiv:2410.06940, 2024.

---

> ### Author Rebuttal · Authors · 2026-03-30
>
> We thank all reviewers for their constructive comments.
> The reviewers remark that our method is novel (P6Pt), the motivation is highly compelling (P6Pt, KhQb, TY1i), the evaluation is broad (TY1i), the improvement is effective (P6Pt, KhQb, TY1i, DKvy), the gains are consistent across multiple settings and metrics (TY1i, P6Pt), the ablation is relevant (TY1i), the work addresses a meaningful limitation (KhQb, TY1i), and the paper is clearly written and easy to navigate (TY1i, DKvy).
> We now address the comments.
>
> **W1: Novelty**
>
> We are the first to inject semantic features in generative NVS and find that improvements in semantic scene and image understanding can improve generative models for NVS.
> Different from the techniques for single-image generation, such as REPA, our novelty lies in leveraging and analyzing scene-level semantics specifically for the novel view synthesis task.
> Nonetheless, we provide a quantitative Comparison with REPA in Table 6 of the paper and achieve better results.
>
> ---
> **W2: Comparison with Prompt-Based Conditioning**
>
> Both SEVA and our method already use CLIP features from the input image as conditioning, which provide coarse global semantic priors similar to prompt conditioning. However, our results show that this alone is insufficient for generative NVS. The key difference is that CLIP-based conditioning mainly captures global, language-level semantics, while our injected features provide spatially grounded, multi-view consistent cues directly from the observed views (e.g., warped DINO features). This is particularly important for NVS, which requires not only semantic plausibility, but also multi-view consistency, scene identity preservation, and geometrically coherent completion of unseen views. The gains over the CLIP-based baseline therefore indicate that explicit semantic feature injection offers complementary benefits beyond CLIP-level conditioning.
>
> ---
> **W3: Non-Warping Baseline**
>
> We thank the reviewer for the suggestion. Following this recommendation, we include comparisons with Gen3R (CVPR 2026) [1] in Table 1-4, a recent camera-controlled generative baseline that does not rely on warping. Our method achieves the best performance across both datasets and trajectory regimes, with particularly clear advantages on long trajectories and on the out-of-distribution Tanks-and-Temples dataset. These results provide additional evidence that our semantic conditioning design is effective.
>
> **Table 1.** Results comparison on RealEstate10K — Short Trajectory (80~100)
>
> | Method | FID ↓ | ImQ ↑ | Drift ↓ | RE ↓ | TE ↓ | MEt3R ↓ | PSNR ↑ | SSIM ↑ | LPIPS ↓ |
> |---|---:|---:|---:|---:|---:|---:|---:|---:|---:|
> | Gen3R | 27.121 | 49.850 | 0.368 | 0.315 | 0.035 | **0.065** | 16.125 | 0.583 | 0.368 |
> | Ours  | **22.726** | **62.060** | **0.251** | **0.189** | **0.019** | 0.067 | **18.024** | **0.646** | **0.251** |
>
> **Table 2.** Results comparison on RealEstate10K — Long Trajectory (≥250)
>
> | Method | FID ↓ | ImQ ↑ | Drift ↓ | RE ↓ | TE ↓ | MEt3R ↓ | PSNR ↑ | SSIM ↑ | LPIPS ↓ |
> |---|---:|---:|---:|---:|---:|---:|---:|---:|---:|
> | Gen3R | 52.283 | 43.773 | 0.472 | 1.708 | 0.212 | 0.128 | 12.004 | 0.497 | 0.628 |
> | Ours  | **26.561** | **66.458** | **0.364** | **0.468** | **0.054** | **0.109** | **13.817** | **0.542** | **0.450** |
>
> **Table 3.** Results comparison on Tank and Temples — Short Trajectory (80~100)
>
> | Method | FID ↓ | ImQ ↑ | Drift ↓ | RE ↓ | TE ↓ | MEt3R ↓ | PSNR ↑ | SSIM ↑ | LPIPS ↓ |
> |---|---:|---:|---:|---:|---:|---:|---:|---:|---:|
> | Gen3R | 39.379 | 58.612 | 0.299 | 0.405 | 0.013 | 0.085 | 14.030 | 0.427 | 0.462 |
> | Ours  | **30.749** | **70.985** | **0.219** | **0.197** | **0.003** | **0.085** | **16.240** | **0.495** | **0.275** |
>
> **Table 4.** Results comparison on Tank and Temples — Long Trajectory (≥250)
>
> | Method | FID ↓ | ImQ ↑ | Drift ↓ | RE ↓ | TE ↓ | MEt3R ↓ | PSNR ↑ | SSIM ↑ | LPIPS ↓ |
> |---|---:|---:|---:|---:|---:|---:|---:|---:|---:|
> | Gen3R | 68.809 | 52.247 | 0.405 | 1.784 | 0.033 | 0.147 | 12.131 | 0.381 | 0.588 |
> | Ours  | **44.381** | **69.829** | **0.266** | **0.577** | **0.008** | **0.119** | **13.825** | **0.426** | **0.405** |
>
> [1] 3D Scene Generation Meets Feed-Forward Reconstruction

---

> > ### Author Rebuttal · Reviewer_DKvy · 2026-04-03
> >
> > Thanks for your reponse! I hope that the author could add the above discussions into the revised version. I will keep my original score.

---

> > > ### Author Response · Authors · 2026-04-07
> > >
> > > We thank the reviewer for the positive feedback and will incorporate the above discussions into the revised version.

---

### Official Review · Reviewer_TY1i · 2026-03-16

**Soundness:** 3
**Presentation:** 3
**Significance:** 3
**Originality:** 3
**Overall Recommendation:** 4
**Confidence:** 3

**Summary:**

This paper proposes SemanticNVS, a camera-conditioned multi-view diffusion approach for novel view synthesis that augments generation with pretrained semantic features. The method builds on a strong NVS backbone and introduces two complementary semantic mechanisms: warped semantic features from source views and iterative semantic conditioning from intermediate denoised predictions during sampling. The intuition is that better semantic understanding of both the input and the generated content should reduce implausible completions under long-range camera motion. Experiments on multiple datasets and trajectory settings show consistent gains in image quality, geometric consistency, and reconstruction quality, especially for longer and more challenging trajectories.

**Compliance With Llm Reviewing Policy:**

Affirmed.

**Key Questions For Authors:**

1. What is the inference-time overhead of SemanticNVS relative to the base model, including DINO extraction and semantic warping?
2. Can the authors provide a more systematic analysis of failure cases where semantic conditioning hurts?
3. How much of the end-to-end gain comes from improving anchor views versus improving later interpolation stages?
4. Could a lighter distilled semantic encoder recover most of the gain at lower cost?

**Limitations:**

Yes

**Strengths And Weaknesses:**

### Strength
The evaluation is broad, the ablations are relevant, and the gains appear consistent across multiple settings.

The paper is clearly written and easy to navigate. The two semantic-conditioning mechanisms are explained well, and the experiments match the central claim closely.

Long-range NVS remains challenging, and semantically implausible hallucinations under larger viewpoint changes are an important problem. The paper addresses a meaningful limitation of current methods, and the reported gains seem practically useful for downstream 3D generation and reconstruction pipelines.

The high-level intuition that stronger semantic conditioning should help generative NVS is natural. Still, the specific integration of warped semantic features and iterative denoising-time semantic feedback is a solid contribution, and the paper does a good job validating where these signals help.

### Weakness
The paper does not quantify the computational overhead of semantic feature extraction and geometric warping clearly enough. Since the method relies on strong external pretrained models and a geometry pipeline, it is important to know how much of the improvement comes at additional cost and complexity.

---

> ### Author Rebuttal · Authors · 2026-03-30
>
> We thank all reviewers for their constructive comments.
> The reviewers remark that our method is novel (P6Pt), the motivation is highly compelling (P6Pt, KhQb, TY1i), the evaluation is broad (TY1i), the improvement is effective (P6Pt, KhQb, TY1i, DKvy), the gains are consistent across multiple settings and metrics (TY1i, P6Pt), the ablation is relevant (TY1i), the work addresses a meaningful limitation (KhQb, TY1i), and the paper is clearly written and easy to navigate (TY1i, DKvy).
> We now address the comments.
>
> **Q1: Computational Overhead**
>
> We report the average inference time and peak memory in Table 1, measured on an H100 over 10 videos with more than 250 frames each. Compared with the baseline (+Warped RGB), warped DINO and iterative DINO deliver significant performance gains with similar runtime (1.303 vs 1.404 s/img). The extra computation from DINO extraction and warping is limited because these operations are performed only during the first-stage generation. We also test the memory and observe only a minor increase from 11.466 GB to 12.290 GB in peak memory. The memory usage of DINO extraction and warping is temporary and does not accumulate across the pipeline. The small memory increase mainly comes from the DINO encoder and the higher diffusion-forward cost introduced by semantic conditioning.
>
> **Table 1.** Computation Cost Comparison.
> | Method | FID ↓ | Infer Time (s/img) ↓ | Mem (GB) ↓ |
> |---|:---:|:---:|:---:|
> | SEVA finetuned | 31.896 | 1.301 | 11.196 |
> | + Warped RGB | 29.942 | 1.303 | 11.466 |
> | + Warped DINO | 28.908 | 1.303 | 12.133 |
> | + Iterative DINO | 26.561 | 1.404 | 12.290 |
>
>
> ---
> **Q2: Failure Cases**
>
> In our observations, semantic conditioning improves performance overall. However, some failure cases are inherited from SEVA. For example, the model may fail on scenes involving humans, animals, or dynamic textures, which may stem from limitations in the scope of its training data. We will clarify this limitation in the main paper.
>
> ---
> **Q3: Source of End-to-End Gain**
>
> The end-to-end gain comes from improving the anchor views. We only apply the proposed semantic conditioning to anchor generation (L267-273), and the improved anchor views subsequently lead to better interpolated views. To verify the source of improvement, we use our method only for anchor-view generation and replace the interpolation stage with SEVA. This yields similar performance, with FID of 21.90 versus 21.84 for the full model. This confirms that the observed end-to-end gain is primarily attributable to the improved anchor views.
>
> ---
> **Q4: Lighter Encoder**
>
> Table 2 reports FID, average inference time, and peak memory for different DINOv2 encoders. We use the ViT-L in our method. Smaller encoders provide weaker gains, while inference time and memory stay comparable. This is because DINO feature extraction is only used in the first-stage generation, so its reduced cost has limited impact on total inference time. Moreover, all models are loaded in float16, making the memory differences modest. Irrespective of the used DINOv2 ViT variant, our method outperforms the SEVA baseline significantly.
>
> **Table 2.** Computation Cost Comparison of Different DINOv2 Encoders.
> | Method | FID ↓ | Infer Time (s/img) ↓ | Mem (GB) ↓ |
> |:---|:---:|:---:|:---:|
> | SEVA finetuned | 31.896 | 1.301 | 11.196 |
> | Ours w/ DINO ViT-S | 28.483 | 1.396 | 11.714 |
> | Ours w/ DINO ViT-B | 27.516 | 1.399 | 11.864 |
> | Ours w/ DINO ViT-L | 26.561 | 1.404 | 12.290 |

---

> > ### Author Rebuttal · Reviewer_TY1i · 2026-04-05
> >
> > Thanks for rebuttal. My concerns have been addressed.

---

> > > ### Author Response · Authors · 2026-04-07
> > >
> > > We thank the reviewer for the positive feedback and will incorporate the above discussions into the revised version.
> > > Since the concerns are resolved, we would also sincerely appreciate it if you would consider raising the score.
> > > Thank you again for the constructive feedback.

---

### Decision · Program_Chairs · 2026-04-30

**Decision:**

Accept (regular)

**Comment:**

This submission received 4x weak accept reviews initially. The rebuttal addressed most of the concerns, and all reviewers maintained their positive scores. Overall, the Area Chairs agree with the reviewers that the paper presents an effective solution for the challenging task of long-range NVS. The extensive evaluations and comparisons including the additional results provided in the rebuttal confirm the effectiveness of the proposed method. The ACs recommend accepting the submission and encourage the authors to incorporate the feedback into the revision.